# The Enzyme 15-Hydroxyprostaglandin Dehydrogenase Inhibits a Shift to the Mesenchymal Pattern of Trophoblasts and Decidual Stromal Cells Accompanied by Prostaglandin Transporter in Preeclampsia

**DOI:** 10.3390/ijms24065111

**Published:** 2023-03-07

**Authors:** Huiyuan Pang, Di Lei, Tingting Chen, Yujie Liu, Cuifang Fan

**Affiliations:** Department of Obstetrics and Gynecology, Renmin Hospital of Wuhan University, Wuhan 430060, China

**Keywords:** preeclampsia, placenta, EMT, decidua, MET, 15-PGDH, PGT

## Abstract

Preeclampsia (PE) is a pregnancy complication beginning after 20 weeks of pregnancy that involves high blood pressure (systolic > 140 mmHg or diastolic > 90 mmHg), with or without proteinuria. Insufficient trophoblast invasion and abnormal decidualization are involved in PE development. However, whether unhealthy placenta and decidua have the same biological activities is unclear. The enzyme 15-hydroxyprostaglandin dehydrogenase (15-PGDH; encoded by *HPGD*) degrades prostaglandin, and prostaglandin transporter (PGT), as a candidate molecule of prostaglandin carriers, helps transport prostaglandin into cells. Whether 15-PGDH and PGT are involved in PE has not been researched. In this study, we investigated the shared pathogenesis of foetal placenta and maternal decidua from the perspective of epithelial–mesenchymal transition (EMT)/mesenchymal–epithelial transition (MET) and explored the combined effects of 15-PGDH and PGT on the EMT/MET of trophoblasts and decidual stromal cells (DSCs). Here, we demonstrated that placental development and decidualization both involved EMT/MET. In PE, both trophoblasts and DSCs show more epithelial patterns. Moreover, 15-PGDH expression was downregulated in the placentas but upregulated in the deciduas of PE patients. Inhibiting 15-PGDH promotes a shift to a mesenchymal pattern of trophoblasts and DSCs depending on the PGT-mediated transport of prostaglandin E2 (PGE2). In conclusion, our results showed that inhibiting 15-PGDH promotes a shift to the mesenchymal pattern of trophoblasts and DSCs and may provide a new and alternative therapy for the treatment of PE.

## 1. Introduction

Development of the maternal–foetal interface is the basis of healthy pregnancy, which depends heavily on placental development and decidualization. The pathogenesis of PE is caused by inhibition of the coordinated development of the maternal–foetal interface. Previous studies have mostly focused on the decidualization of DSCs [1,2] or the invasion of trophoblasts separately [3]; however, since half of a foetus’s genes originate from the mother, a few studies have focused on the underlying link between decidualization and placental development. The processes of trophoblast differentiation and decidualization are both involved in EMT/MET [4,5,6]. During placental development, trophoblasts undergo continuous differentiation. Cytotrophoblasts (CTBs) differentiate into extravillous trophoblasts (EVTs) which invade the decidua. The process of differentiation from CTBs to EVTs involves EMT [4]. DSCs contained in decidual tissue undergoing decidualization do not depend on the implantation of human embryos in each menstrual cycle. The process of decidualization involves MET [6]. However, as a common biological behaviour of placental development and decidualization, the shared pathology of trophoblast EMT and DSC MET in PE has seldom been researched.

15-PGDH is a key factor in the degradation of PGE2, and inhibiting 15-PGDH impairs the degradation of PGE2 [7]. Embryonic development depends on the appropriate prostaglandin concentration [8]. Concentrations of prostaglandins that are too high or too low will inhibit embryo implantation. The metabolic regulation of PGs is very important. Most previous studies have focused on the roles of COX-2, a key enzyme in the synthase of prostaglandins, but there have been few studies on 15-PGDH, another important member of the enzyme family that regulates PGE2. Recently, 15-PGDH was found to be closely related to cancer invasion and tissue regeneration [7,9,10]. The decomposition of PGE2 relies on PGT-mediated transport [11]. PGT is a transporter with 12 transmembrane domains and a lactate-PG transport mechanism [12]. However, there is no study about how 15-PGDH and PGT cooperate and affect trophoblast and DSC EMT/MET in PE.

In this study, the abnormal placental development and decidualization in PE were both related to the shift between epithelial and mesenchymal patterns, which helps with the exploration of therapeutic targets of PE. We mainly studied the role of 15-PGDH in PE. We found that 15-PGDH protein expression displays opposite patterns in the foetal placenta and maternal decidua in normal pregnancy, and this expression is altered in PE. We then explored the effect of 15-PGDH on the biological behaviour of trophoblasts and DSCs. Furthermore, we explored the cooperative effect of 15-PGDH and PGT in trophoblasts and DSCs.

## 2. Results

### 2.1. Partial EMT Occurs during Placental Development

In the first-trimester villi, we performed coimmunofluorescence to demonstrate that the development of CTBs into EVTs involved partial EMT. Human leukocyte antigen (HLA-G) is gradually expressed when CTB develops into EVT, where it mediates maternal–foetal immune tolerance during pregnancy [13]. HLA-G is a biomarker of CTB development. Since HLA-G is specifically expressed on EVTs in the placenta, HLA-G (pink in Figure 1A, green in Figure 1C) specifically represents EVTs, and the adjacent region represents CTBs and syncytiotrophoblasts (STBs), which are formed by the cell–cell fusion of CTBs. We compared the expression of cadherin-1 (CDH1) and cadherin-2 (CDH2) in HLA-G^+^ EVT with HLA-G^−^ CTB to determine how EMT actually occurs in CTB development (Figure 1B,D). CDH1 (red) protein expression decreased during HLA-G^-^ CTB development into HLA-G^+^ EVT (Figure 1A,B), while CDH2 (red) did not show a significant difference between CTB and EVT (Figure 1C,D). Therefore, when CTBs develop into EVTs, CTBs gradually lose their epithelial pattern but do not gain their mesenchymal pattern significantly, which demonstrates that partial EMT occurs during trophoblast development.

### 2.2. MET Occurs during Decidualization

Human decidual stromal cells (hESCs) were induced to decidualize (decidualized hESC: dhESC) according to the above methods [14]. Western blots revealed that prolactin (PRL) and insulin-like growth factor binding protein 1 (IGFBP1), the traditional markers of decidualization, increased (Figure 1E), indicating that in vitro hESC decidualization was successful; moreover, CDH1 increased, and vimentin decreased during hESC decidualization in vitro (Figure 1E,G). Phalloidin-specific binding of F-actin further demonstrated that during decidualization, hESCs gradually lost fibroblastic structural characteristics and gained typical polygonal morphology (the shape change of hESCs is labelled with arrows in Figure 1F). During decidualization, hESCs undergo MET-like changes.

### 2.3. Insufficient EMT Changes in the Placenta and Excess MET-Like Changes in Decidual Tissue in Preeclampsia

We collected placentas and decidua specimens and performed immunohistochemical staining. Immunohistochemical staining showed CDH1 and CDH2 changes in trophoblasts but not in other cells and revealed that CDH1 protein expression was increased in the placentas of the PE group compared to those of the normal group (*n* = 6; *p* ≤ 0.01) (Figure 2A,B,E), while CDH2 was decreased in the placenta of the PE group compared to the normal group (*n* = 6; *p* ≤ 0.01) (Figure 2C–E).

We also demonstrated that abnormal decidua in PE is accompanied by excessive MET (Figure 3). In this study, we only focused on DSCs, which are the exact cells that undergo decidualization. Therefore, we also identified DSCs specifically. Since the decidual composition is complex and includes DSCs, trophoblasts and other cells including immune cells, we first identified CK7^−^vimentin^+^ DSCs [2] (Cytokeratin 7: CK7), (CK7: green; vimentin: red) by coimmunofluorescence before further study (Figure 3A). Vimentin helped determine that the larger and rounder cells were DSCs by immunohistochemistry (Figure 3B). Therefore, in further research, we will only focus on these DSCs (the area of DSCs is labelled with arrows, and non-DSCs are labelled with triangles in Figure 3C,E). CDH1 protein expression was increased in the DSCs of the PE group compared to those of the normal group (*n* = 6, *p* ≤ 0.05). In contrast, CDH2 and vimentin were decreased in the DSCs of the PE group compared to those of the normal group (*n* = 6, *p* ≤ 0.05) (Figure 3C−E). We demonstrated that abnormal decidualization in PE (Appendix A) is accompanied by excessive MET.

### 2.4. Expression of 15-PGDH Is Downregulated in the Placenta but Upregulated in the DSCs of PE Patients

Immunohistochemistry revealed that 15-PGDH was downregulated in trophoblasts in PE patients (Figure 4A,E) (*n* = 6; *p* ≤ 0.05). Immunohistochemical staining showed that 15-PGDH was upregulated in the DSCs of patients with PE (Figure 4B,F) (*n* = 6; *p* ≤ 0.05). Therefore, 15-PGDH is involved in both placental and DSC abnormalities in PE.

### 2.5. Location of 15-PGDH in the Villi and DSCs

To explore the location of 15-PGDH in trophoblasts, we used first-trimester villi to detect 15-PGDH location. CTB development is accompanied by gradual HLA-G expression and invasion into the maternal decidua. Therefore, in the first-trimester villi, we can clearly identify CTBs and EVTs to explore the 15-PGDH expression pattern in these two kinds of cells. Thus, how 15-PGDH expression changes during CTB development can be explored. Therefore, we first used villous coimmunofluorescence to reveal that 15-PGDH expression increased significantly during the development of CTBs into EVTs (the area of CTBs is labelled with **+**, and the area of EVTs is labelled with triangles), which also demonstrated that 15-PGDH was specifically located in trophoblasts instead of the stroma and was closely linked to trophoblast EMT (CK7: green; 15-PGDH: red) (Figure 4C). CK7 is a marker of trophoblasts [15].

To explore the location of 15-PGDH in the decidua, we used decidual tissue obtained during caesarean section since decidualization is not accompanied by DSC location changes. In decidual tissue, coimmunofluorescence revealed that 15-PGDH was located in the cytoplasm of the DSCs (vimentin: green; 15-PGDH: red) (Figure 4D).

### 2.6. SW033291 Inhibits 15-PGDH and Upregulates PGE2 Expression

SW033291, a 15-PGDH inhibitor, was used to treat Jeg3 cells and hESCs at different levels (0, 50, 100, 200, 350 and 500 nM) to detect the relationship between SW033291 concentration and PGE2 concentration. Then, cell lysates were used to perform ELISA to determine PGE2 concentrations. The PGE2 concentration increased with SW033291 in a dose-dependent manner (Figure 5A). Therefore, we chose these concentrations of SW033291 for further research.

### 2.7. Inhibiting 15-PGDH Promotes a Shift to a Mesenchymal Pattern in Trophoblasts and DSCs

To explore the function of 15-PDGH in trophoblasts and DSCs, we added SW033291 (0, 50, 100, 200, 350, and 500 nM) to the culture medium of Jeg3 cells and dhESCs, and Western blotting was used to reveal the expression of EMT/MET markers, decidualization markers (PRL, IGFBP1) and the CTB development marker (HLA-G). After the addition of SW033291, CDH1 decreased while CDH2 increased in dhESCs in a dose-dependent manner, as shown in Figure 5F. We chose the trophoblast Jeg3 cell line to test this effect (Figure 5D,E) since Jeg3 is the only trophoblast cell line that expresses HLA-G. HLA-G did not show a significant change during the EMT of trophoblasts promoted by SW033291 (Figure 5E). In dhESCs, PRL and IGFBP1 also increased in a dose-dependent manner (Figure 5F,G). Among them, PRL expression increased after exposure to SW033291, while IGFBP1 expression increased significantly at 500 nM (Figure 5F,G).

To demonstrate the role of 15-PGDH, we used F-actin staining to evaluate cytoskeletal reorganization, another important characteristic of EMT/MET. Compared with those of the normal control group, HTR8 and dhESCs treated with SW033291 (500 nM) showed fibroblastic-like shapes, which are characterized by strong F-actin stress fibres arranged longitudinally through the major axis (Figure 5B,C).

### 2.8. PGT Is Upregulated When 15-PGDH Is Inhibited

To explore the effect of 15-PGDH on PGT, we performed Western blotting to determine how PGT expression changed when 15-PGDH was inhibited. Western blot analysis showed that in both Jeg3 cells and hESCs, PGT expression was upregulated when cells were treated with SW033291, as shown in Figure 6A,B, which showed that the inhibition of 15-PGDH promotes the mesenchymal pattern involving PGT.

### 2.9. PGT Is Upregulated in the Placenta but Downregulated in the DSCs of PE Patients

Immunohistochemistry also revealed that PGT was upregulated in the placentas of the PE group (Figure 6C,E) (*n* = 6; *p* ≤ 0.05) but downregulated in the decidua of the PE group (Figure 6D,F) (*n* = 6; *p* ≤ 0.05). Therefore, PGT expression changed along with 15-PGDH both in vitro and in vivo.

### 2.10. PGT Coexpression with 15-PGDH in the Villous and DSCs

To further determine the possible synergistic role of 15-PGDH and PGT, we conducted coimmunofluorescence and revealed that 15-PGDH and PGT were both located in trophoblasts (Figure 6G) and DSCs (Figure 6H). Coimmunofluorescence was performed to reveal that PGT was located in trophoblasts (Appendix A) and DSCs (Appendix A). However, how 15-PGDH and PGT affect each other requires further study.

### 2.11. Inhibition of 15-PGDH Promotes the Mesenchymal Pattern Depending on the Normal Transport Function of PGT

To further demonstrate the relationship between 15-PGDH and PGT, we treated Jeg3, HTR8 and dhESCs with both SW033291 and indocyanine green (ICG), a PGT inhibitor.

We first performed ELISA to detect PGE2 in the cell lysates of Jeg3 cells and dhESCs from three groups: the NC, SW033291 and SW033291 + ICG groups. Both PGE2 in Jeg3 cells and dhESC cells in the SW033291 group were significantly increased; however, when treated with ICG, the concentration of PGE2 then decreased, which demonstrated that the PGE2 increase depended on the lactate-PG transport mechanism of PGT (Figure 7A,B).

We also performed Western blotting to reveal changes in the expression of related molecules. First, in both Jeg3 cells and dhESCs, SW033291 promoted mesenchymal markers, such as CDH2, while inhibiting the epithelial marker CDH1; however, this effect was counterbalanced by ICG (67 μM) (Figure 7E−H). Second, in Jeg3 cells, HLA-G still did not show a significant change when Jeg3 cells transitioned between an epithelial or mesenchymal pattern, indicating that the concentration of PGE2 and the expression of 15-PGDH have no effect on HLA-G (Figure 7G,H). In regard to PRL expression, ICG can counterbalance exposure to SW033291 (Figure 7E,F).

To corroborate these effects, we also used F-actin staining to evaluate cytoskeletal reorganization (Figure 7C,D). HTR8 (Figure 7D) and dhESCs (Figure 7C) treated with SW033291 (500 nM) showed fibroblastic-like shapes, while HTR8 and dhESCs treated with SW033291 (500 nM) and ICG (67 μM) exhibited a change in the localization of actin to the edge of the cell membrane, which represents a typical polygonal morphology (Figure 7C,D). ICG can reverse the effect of SW033291. PGT serves as an exchange for PGs with lactate. When PGT is inhibited by ICG and cells cannot take up PGE2, even inhibiting 15-PGDH is still not useful to promote the mesenchymal pattern.

### 2.12. Preeclampsia Rat Model Construction and Expression Detection of 15-PGDH and PGT

Given clinical ethics, we could not obtain a complete maternal–foetal interface in patients with PE. However, in the animal model, we obtained the complete maternal–foetal interface as described above. We successfully constructed two kinds of PE rat models according to the above methods. The blood pressure of the reduced uterine perfusion pressure (RUPP) group [16] and NG-nitroarginine methyl ester hydrochloride (L-NAME) group increased significantly (approximately 20 mmHg) (Figure 8A,B). The CK7^−^vimentin^+^ zone is consistent with decidual cells, and CK7^+^ indicates the labyrinth zone, which is consistent with the placentas of humans. HE staining showed that the structure of the maternal–foetal interface in the L-NAME and RUPP groups was significantly changed. First, the placenta was degenerated (Figure 8E,G), the sinus-like structure in the decidua layer became larger, and the number of cells was reduced (Figure 8E,F). Second, the uterine volume was reduced (Appendix A), the growth and development of foetal rats in the RUPP group were restricted and the growth and development in the L-NAME group were slightly restricted (Appendix A). Moreover, HE staining revealed that the structure of the kidney was damaged as well, with widening visible glomerular cysts and weakening of the glomerulus (Appendix A, glomerulus labelled with triangles).

Immunohistochemistry demonstrated that 15-PGDH was significantly downregulated in the placenta (Figure 9B) (*n* = 6; *p* ≤ 0.05) but upregulated in the decidua of the RUPP rat model and L-NAME PE rat model group (Figure 9A) (*n* = 6; *p* ≤ 0.05). PGT was significantly upregulated in the placenta (Figure 9D) (*n* = 6; *p* ≤ 0.05) but downregulated in the decidua of the RUPP rat model and L-NAME PE rat model (Figure 9C) (*n* = 6; *p* ≤ 0.05). Figure 9E represents negative control of placenta and decidua.

## 3. Discussion

In our study, we verified EMT/MET during placental development and decidualization. Partial EMT occurs during trophoblast differentiation, and MET occurs during decidualization. Then, we found that both the trophoblasts and DSCs of PE patients tended to have more epithelial patterns, indicating insufficient EMT of trophoblasts and excessive MET during decidualization of DSCs. 15-PGDH was differentially expressed in the placenta and decidua of PE patients. Inhibiting 15-PGDH promoted trophoblast differentiation and DSC decidualization and led to a shift to a mesenchymal pattern in both groups in a dose-dependent manner. Inhibiting 15-PGDH can upregulate PGT, which can lead to the uptake of more PGE2. Inhibiting 15-PGDH promotes a mesenchymal pattern depending on the lactate-PG transport mechanism of PGT.

EMT/MET was once thought to be a transition between a complete epithelial pattern or mesenchymal pattern. However, with more research, EMT/MET should be further elucidated. EMTs/METs are multistep, reversible, dynamic biological processes of cell differentiation and dedifferentiation, with cells transitioning along various stages, including various partial EMT states, which are also characterized by cytoskeleton and molecular marker changes. In regard to trophoblast development, in our study, CDH1 was decreased in the process of CTB EMT, while CDH2 was not significantly increased, which was consistent with some previous dissertations. In 2015, a study found that EMT occurring in placental development lacked traditional characteristics of EMT types 1–3 and defined trophoblast EMT as type 0 [17]. A review published in 2019 in BMJ defined trophoblast EMT as partial EMT [18]. In our research, we used coimmunofluorescence to reveal partial EMT during placental development.

Whether EMT/MET is the basis for cells to undertake different biological functions and cell differentiation, is parallel to cell differentiation, or only prepares for invasion or metastasis is an interesting question. HLA-G is expressed during CTB EMT and is a marker of CTB development, which is a kind of human leukocyte antigen that plays a major role in mediating immune tolerance. This molecule can prevent the embryo from being attacked by immune cells in the decidua. The expression of HLA-G is decreased in patients with PE [19], which is consistent with the decrease in EMT in patients with PE. However, in our research, we found that HLA-G expression did not significantly change when trophoblasts changed along the epithelial and mesenchymal spectra. In addition, the blots of CDH1 show numerous panels, which may be caused by alternative splicing or protein modification. Whether modification of CDH1 involves new mechanism of EMT/MET needs further research.

In the process of decidualization, MET occurs with PRL and IGFBP1 expression. In DSCs in PE patients, excess MET is accompanied by PRL and IGFBP1 deficiency; in an in vitro cell model, insufficient MET was accompanied by PRL and IGFBP1 increases. These phenomena seem contrary, which may be due to the oversimplification of EMT/MET. The relationship between EMT/MET and decidualization is complex. A review of EMT/MET in 2016 may help explain these phenomena [20]. The authors of this review predicted and described hypothetical EMT/MET transitional states, among which there are several special states, including intermediate state (EM) 1 and EM3, indicating that cells are on more thermodynamic peaks with more metastable states than complete epithelial or mesenchymal patterns. Between EM1 and EM3, EM2 has a higher mesenchymal/epithelial score (M/E score) than EM1 and a lower score than EM3 but with lower energy, which means EM2 is more stable than EM1 [20]. Therefore, we predict that the “fitting curve” between EMT/MET and IGFBP1 and PRL is a “sinusoidal curve” instead of a “linear regression curve”, which means that IGFBP1 and PRL do not increase with increasing M/E scores. Among the decidualization procedures, there may be different stages with different PRL or IGFBP1 expression or M/E scores. If we understand EMT/MET in this way, many seemingly contradictory conclusions are reasonable. However, due to the limitations of the experimental model, we could not build decidualization with multiple partial EMT states to detect the relationship between EMT/MET. Whether EMT/MET is the basis of HLA-G, IGFBP1, or PRL expression or a parallel occurrence still deserves further study.

The placentas and deciduas of patients with PE showed an excessive epithelial pattern and insufficient mesenchymal pattern, indicating that EMT of the placenta is insufficient, while MET of the decidua is excessive. In the mesenchymal pattern, the loss of stress fibres from the centre of the cell body promotes the movement of cells and makes it easier for trophoblasts and decidual cells to invade each other and for an embryo to attach to the mother, completing vascular remodelling and forming a healthy maternal–foetal interface. This phenomenon may be one of the reasons for the shallow implantation of trophoblasts. Our experiment proved that the epithelial pattern in the deciduas of PE patients was excessive, which was consistent with another study [21]. However, studies have shown that the deciduas of recurrent spontaneous abortion (RSA) show a shift to an excess mesenchymal pattern [22]. Previous studies have shown that RSA and PE share a similar pathogenesis; however, from the perspective of decidua MET, there are essential differences between the decidualization abnormalities of RSA and PE patients. However, this conclusion still needs multicentre and multisource confirmation.

Prostaglandins play an important role in embryo implantation, formation of the maternal–foetal interface and initiation of labour. Prostaglandins regulate embryo implantation at lower concentrations and inhibit embryo implantation at higher concentrations. Therefore, the precise regulation of prostaglandin metabolism is very important for the formation of the maternal–foetal interface. Aspirin, as an inhibitor of prostaglandin synthase COX-2, has been widely used in the clinic to prevent PE [23,24]. PGT and 15-PGDH also play important roles in regulating prostaglandin concentrations. PGT and 15-PGDH may also become targets for the treatment of PE, similar to aspirin.

However, the expression of 15-PGDH is increased in the placentas of PE patients, but it promotes the epithelial–mesenchymal transformation of trophoblasts. This conclusion seems contradictory, and the possible reasons are listed. 1. Lesions exist in early pregnancy in PE patients; however, for ethical reasons, research on PE relies on the placenta at the time of delivery. Therefore, 15-PGDH may be compensated in a long gestational duration, or the expression of 15-PGDH may be affected by aspirin, which is commonly used. 2. The other explanation involves a more controversial viewpoint. Is the placenta an organ that causes PE or an injured organ that is harmed by PE [1,25]? This study tested the hypothesis that decidual defects are an important determinant of the placental phenotype [1]. In our study, we demonstrated that 15-PGDH, which inhibits mesenchymal patterns and decidualization, is upregulated in the deciduas but downregulated in the placentas of PE patients. This conclusion suggests that the decidua and the microenvironment of trophoblasts are potential causes of PE. However, testing only one molecule is not enough, and this issue requires further research.

## 4. Materials and Methods

### 4.1. Patients and Sample Collection

First trimester placental villi were obtained from healthy women undergoing elective surgical termination of their pregnancies from 6–8 weeks of gestation. A total of 16 placental tissue samples from patients with PE and matched healthy controls were collected in the Obstetrics and Gynaecology Department, Renmin Hospital of Wuhan University (Wuhan, China) from March 2022 to May 2022 (Appendix A), and informed consent was obtained from all the patients in advance. Placental tissues and decidual tissues were collected from women in the third trimester during caesarean section. The placenta and decidua specimens were washed with sterile PBS, fixed in 4% paraformaldehyde, or quick-frozen in liquid nitrogen for later use. Human sample collection was authorized by the Ethical Review Board of Renmin Hospital, Wuhan University (WDRY2021-K177) and performed in accordance with the Declaration of Helsinki.

### 4.2. Cell Culture and Differentiation

Human trophoblast cell lines (Jeg3, HTR8) and human decidual stromal cell line (hESC) were purchased from the Cell Bank of the Chinese Academy of Sciences (Shanghai, China). Jeg3 and HTR8 cells were cultured in a 5% humidified carbon dioxide atmosphere at 37 °C in Dulbecco’s modified Eagle’s medium (DMEM)/F-12 (Gibco, Life Technologies, Grand Island, NY, USA) with 10% foetal bovine serum (Gibco, Life Technologies, Grand Island, NY, USA), 50 mg/mL streptomycin, and 50 U/mL penicillin. Jeg3 is the only trophoblast cell line that expresses human leukocyte antigen G (HLA-G), which is gradually expressed during development from CTBs to EVTs.

hESCs were cultured in a 5% humidified carbon dioxide atmosphere at 37 °C in phenol red-free Dulbecco’s modified Eagle’s medium (DMEM)/F-12 (Meilunbio, Dalian, China) with 10% foetal bovine serum, 50 mg/mL streptomycin, and 50 U/mL penicillin. One micromolar medroxyprogesterone-17-acetate (MPA) (HY-B0469, MedChemExpress, NJ, USA) and 0.5 mM N6,20-O-dibutyryladenosine cAMP sodium salt (db-cAMP) (HY-B0764, MedChemExpress, Monmouth Junction, NJ, USA) were added to the culture for 6 days to induce hESC decidualization in vitro [14].

### 4.3. SW033291 and ICG Treatment

Cells were treated with 500 nM SW033291 (HY-16968, MedChemExpress, Monmouth Junction, NJ, USA) [8] to inhibit 15-PGDH or with 67 μM indocyanine green (ICG) (HY-D0711, MedChemExpress, Monmouth Junction, NJ, USA) to inhibit PGT.

### 4.4. Western Blot Analysis

Total protein was extracted from cells with RIPA buffer, PMSF protease inhibitors, and a phosphatase inhibitor (all from Servicebio, Wuhan, China) and then ultrasonicated and centrifuged at 12,000  rpm for 10  min at 4 °C. The supernatants, fixed with loading buffer (Elabscience, Wuhan, China), were heated for 5 min at 100 °C and then kept at −20 °C. Protein from each sample was resolved through 10% SDS‒PAGE (PG212 Omni-EastTM, EpiZyme, Shanghai, China) and then transferred to polypropylene difluoride membranes (Millipore, USA) for 30 min (PS108P, EpiZyme, Shanghai, China). The membranes were blocked in blocking buffer (PS108P, EpiZyme, Shanghai, China) for 15 min at room temperature and then immunoblotted with primary antibodies against 15-PGDH (11035-1-AP, Proteintech, Wuhan, China), PGT (ab150788, Abcam, Cambridge, UK), E-cadherin, also named CDH1, (20874-1-AP, Proteintech, Wuhan, China), N-cadherin, also named CDH2, (22018-1-AP, Proteintech, Wuhan, China), vimentin (10366-1-AP, Proteintech, Wuhan, China), IGFBP1 (Ab-DF7130, Affinity, Jiangsu, China), prolactin/PRL (Ab-DF6506, Affinity, Jiangsu, China), HLA-G (66447-1-Ig, Proteintech, Wuhan, China), and GAPDH (10494-1-AP, 1:5000, Proteintech, Wuhan, China) overnight at 4 °C, followed by incubation with goat anti-rabbit IgG (H  +  L) or goat anti-mouse IgG (H  +  L) (GB23303, 1:1000, Servicebio Technology Co., Wuhan, China) for 1.5 h at 4 °C. Protein expression was detected by a chemiluminescent detection system (Bio-Rad, Hercules, CA, USA) using ECL Plus reagents (Servicebio Technology Co., Wuhan, China). The expression levels of targeted proteins were normalized to GAPDH. Western blot analysis was conducted using ImageJ Pro Plus version 6.0 software.

### 4.5. Immunohistochemistry

Paraffin-embedded placental tissues and decidua tissues were sectioned at a thickness of 10 nm, dewaxed, rehydrated and blocked with BSA. Sections were incubated overnight with primary antibodies as described for Western blotting. Sections for negative control were incubated without primary antibody. After washing with phosphate-buffered saline (pH 7.4), the sections were incubated for 2 h with HRP-conjugated secondary antibodies: Alexa Fluor 488-conjugated goat anti-mouse IgG (A32723; Thermo Fisher Scientific, Waltham, MA, USA) and Alexa Fluor 568-conjugated goat anti-rabbit IgG (A11011; Thermo Fisher Scientific, Waltham, MA, USA). Nuclei were visualized with DAPI (Beyotime, Shanghai, China). The digital image processing system ImageJ Pro Plus version 6.0 was then employed to evaluate Area and IntDen of IHC. Average optical density (AOC) = IntDen/area, and AOD was calculated.

After deparaffinization and rehydration, standard H&E staining was performed for morphological analysis.

### 4.6. Immunofluorescence

The steps before incubation with primary antibodies were the same as those used for immunohistochemistry. The samples were incubated with the desired dilutions of primary antibodies overnight at 4 °C. The samples were then incubated with fluorescence-labelled secondary antibody for 1 h (Beyotime, Shanghai, China) at room temperature and counterstained with 4′-6-diamidino-2-phenylindole (DAPI) (Beyotime, Shanghai, China). The primary antibodies used in the study were the same as those used for Western blotting. The secondary antibodies used included anti-rabbit IgG (H + L) Alexa Fluor 555 (Invitrogen, San Diego, CA, USA; A-31572) and anti-goat IgG (H + L) Alexa Fluor Plus 488 (Invitrogen; A32814). A confocal laser scanning microscope (Olympus FV1000, Tokyo, Japan) was used to observe the fluorescence signal. Five visual fields with tissue were selected for analysis. The pixel intensity per unit area was assessed using ImageJ (1.52a, National Institutes of Health, Rockville, MD, USA).

### 4.7. ELISA

The level of PGE2 in cell lysates was identified by ELISA. Ultrasonicated cells were centrifuged at 12,000  rpm for 10  min at 4  °C. PGE2 concentrations in the cells were detected by a PGE2 competitive ELISA kit (EK8103/2, Multi Science, Hangzhou, China). All the abovementioned analyses were performed according to the relevant manufacturer’s instructions.

### 4.8. F-Actin Staining

After treatment with SW033291 or ICG, all round coverslip samples were washed with PBS and fixed in 3.7% paraformaldehyde (Servicebio, Wuhan, China) for 15 min, and the coverslips were washed three times with PBS. Then, the coverslips were permeabilized with 0.1% Triton X-100 for 10 min and stained with rhodamine conjugated to phalloidin (Phalloidin-iFluor 647 Reagent, Abcam, Cambridge, UK) for 30 min at 37 °C. After three washes with PBS, the nuclei were visualized with DAPI (100 nM) for 10 min.

### 4.9. Preeclampsia Rat Model Construction

Twenty-five Sprague–Dawley female rats and fifteen Sprague–Dawley male rats, purchased from Beijing Vital River Laboratory Animal Technology Co., Ltd., China, were used in our studies. After adapting to culture for 7 days in specific pathogen-free (SPF) conditions at Renmin Hospital of Wuhan University, the rats were mated, and the day was recorded as day 0.5 of gestation. Pregnant rats were randomly divided into 4 groups according to their body weight. The L-NAME group was subcutaneously injected with NG-nitroarginine methyl ester hydrochloride (L-NAME) (HY-18729A, MedChemExpress, Monmouth Junction, NJ, USA) from the 10th day of pregnancy to the 18th day of pregnancy (100 mg/kg × day), while the normal pregnancy control group was injected subcutaneously with physiological saline. Blood pressure was measured on gestational day 10, 13, 16 and 18 by noninvasive tail-cuff system (CODA system, Kent Scientific, Torrington, CT, USA). Rats in RUPP group were operated on at 14.5 days of pregnancy. The surgical procedure was performed as described previously [16,26], and the skin was cut along the midline of the abdomen. The omentum was gently pushed with two cotton swabs, and the intestinal tube was pushed with wet gauze to expose the posterior abdominal wall. The abdominal aorta was found, the surrounding fascial tissue was separated, and then, the abdominal aortic silver clip was slid to 0.5 cm above the abdominal aortic bifurcation. Then, the ovarian artery silver clips were placed, as Figure 5C shows (white triangles indicate the position of the silver clips). No silver clip was placed in rats in the sham group. The sham group means negative control group. On gestational day 18, carotid arterial catheters were inserted for blood pressure measurements. After blood pressure measurement, tissues were collected. Collection of the maternal–foetal interface: we cut the uterus along the opposite side of the uterine blood vessels, peeled off the amnion, and cut off the umbilical cord. Without separating the placenta and the uterus, we completely preserved the maternal–foetal interface and maintained its morphology. Kidneys and other organs were also collected. The specimens were washed with sterile PBS and then fixed in 4% paraformaldehyde for later use. All animal studies were approved by the ethics committee for laboratory animal welfare (IACUC) of Renmin Hospital of Wuhan University [No. WDRM animal (f) No. 2022103C].

### 4.10. Statistical Analysis

Statistical significance was determined by SPSS 20.0 software, and *p* = 0.05 was the threshold. Student’s *t* test or one-way ANOVA was used to analyse differences between two or more groups.

## 5. Conclusions

We demonstrate for the first time that abnormal 15-PGDH and PGT expression could be associated with abnormal EMT/MET in patients with preeclampsia. 15-PGDH inhibition improves the mesenchymal pattern of both trophoblasts and decidual stromal cells, which are the most important components of the maternal–foetal interface. Inhibiting 15-PGDH upregulated PGT expression on the cell membrane. 15-PGDH promotes PE relying on PGT, which functions as an electrogenic anion exchanger for PG with lactate. The present study provides new insights into the potential role of 15-PGDH and PGT in PE treatment.

## Figures and Tables

**Figure 1 ijms-24-05111-f001:**
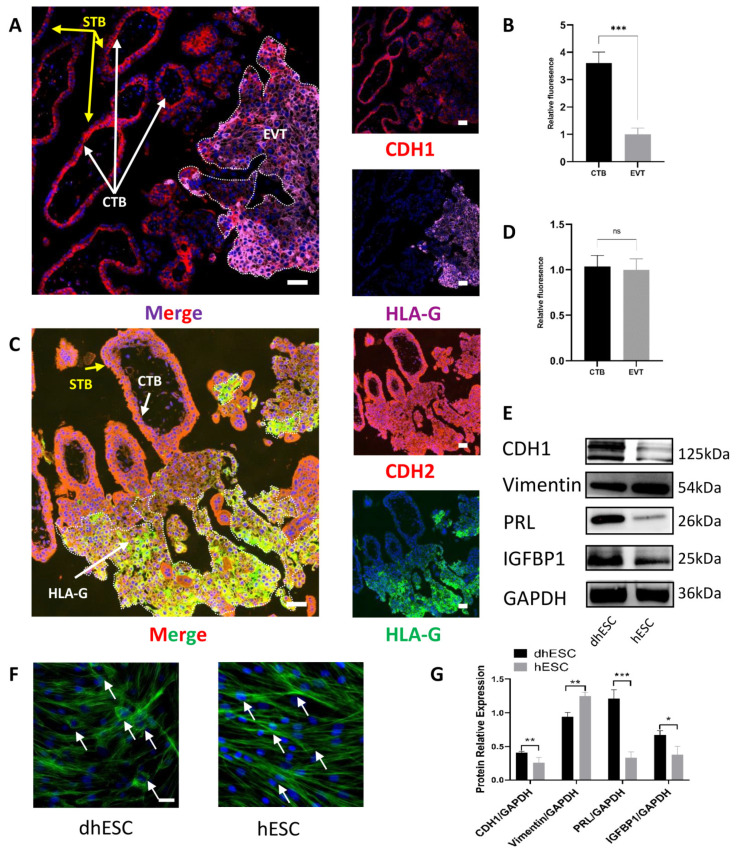
Partial EMT occurs in placental development, while MET occurs during decidualization. (**A**,**B**) CDH1 location and expression levels in CTBs and EVTs were measured by coimmunofluorescence in early pregnancy villi (*n* = 3). Scale bar = 50 nm in (**A**,**B**). (**C**,**D**) CDH2 location and expression levels in CTBs and EVTs were measured by coimmunofluorescence in early pregnancy villi (*n* = 3). Scale bar = 50 nm in (**C**,**D**). *** *p* < 0.001; ns, not significant. (**E**,**G**) Western blot tests of MET and decidualization markers during decidualization in vitro. The experiment was repeated three times independently. dhESC: decidualized hESC. GAPDH served as a loading control. Band intensities were quantified and normalized to the GAPDH values. Values are the mean ± SD. * *p* < 0.05; ** *p* < 0.01; *** *p* < 0.001; ns, not significant. (**F**) F-actin staining of dhESCs and hESCs shows cytoskeletal changes during decidualization. Scale bar = 50 nm. CTB: cytotrophoblast; EVT: extravillous trophoblasts; EMT: epithelial–mesenchymal transition.

**Figure 2 ijms-24-05111-f002:**
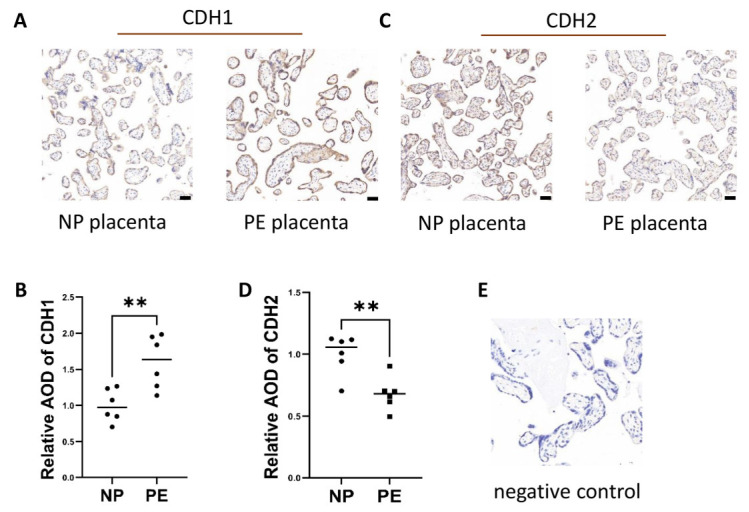
Excess epithelial pattern in trophoblasts in PE patients. (**A**,**B**) CDH1 protein expression in placental tissues from women with NP and PE was measured by immunohistochemical staining. (**C**,**D**) CDH2 protein expression in placental tissues from women with NP and PE was measured by immunohistochemical staining. (**E**) represents negative control. Values are the mean ± SD. ** *p* < 0.01; ns, not significant. *n* = 6 each. Scale bar = 50 nm.

**Figure 3 ijms-24-05111-f003:**
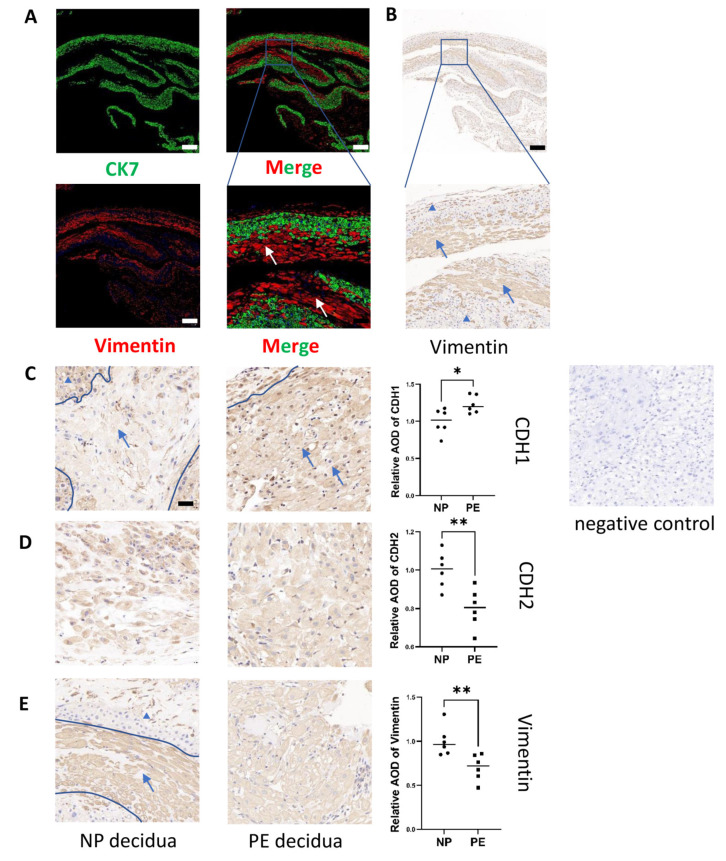
Excess epithelial pattern in DSCs in PE patients. (**A**,**B**) Immunofluorescence and immunohistochemical staining identified DSCs in decidual tissues. Red: vimentin; Green: CK7, scale bar = 500 nm. The experiment was repeated 3 times independently. (**C**,**D**,**E**) Measurement of CDH1, CDH2 and vimentin protein expression in DSCs from women with NP and PE by immunohistochemical staining. The area of DSCs is labelled by arrows and non-DSCs are labelled with triangles, Scale bar = 20 nm. Values are the mean ± SD. * *p* < 0.05; ** *p* < 0.01; ns, not significant. *n* = 6 each. PE: preeclampsia; NP: normal pregnancy; DSC: decidual stromal cell.

**Figure 4 ijms-24-05111-f004:**
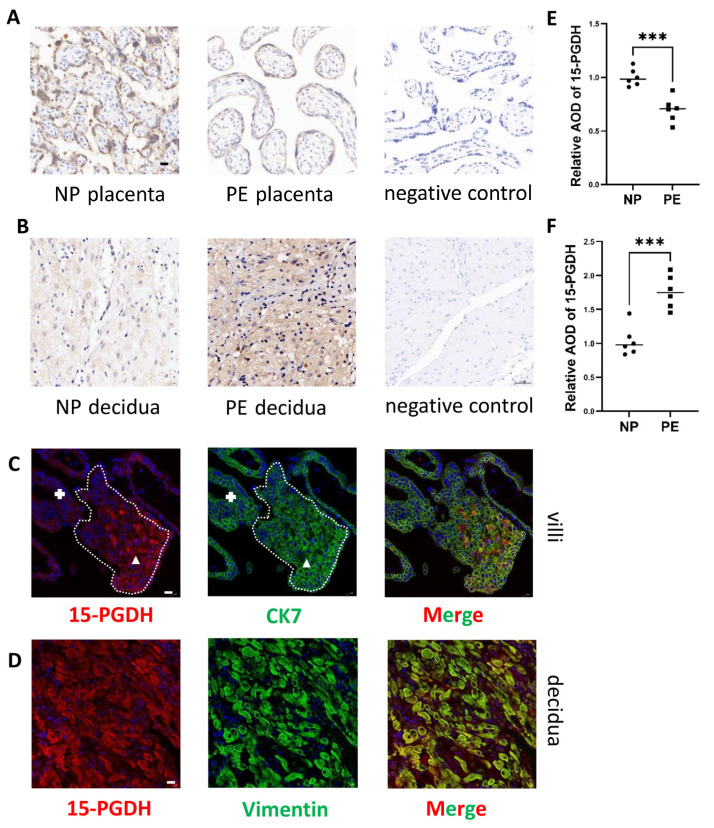
Differential expression of 15-PGDH in PE and the location of 15-PGDH. (**A**,**B**,**E**,**F**) Immunohistochemical staining showed the relative quantification of 15-PGDH in the placenta and decidua of the PE and NP groups. Brown staining represents the target protein. Scale bar = 20 nm, *n* = 6, Values are the mean ± SD. *** *p* < 0.001. *n* = 6 each. (**C**) Differential expression of 15-PGDH in CTBs and EVTs was identified by immunofluorescence, red: 15-PGDH; green: CK7, Bar = 20 nm. The area of CTBs is labelled with **+**, and the area of EVTs is labelled with triangles. (**D**) Specific location of 15-PGDH in DSCs was identified by immunofluorescence, red: 15-PGDH; green: vimentin, Bar = 20 nm.

**Figure 5 ijms-24-05111-f005:**
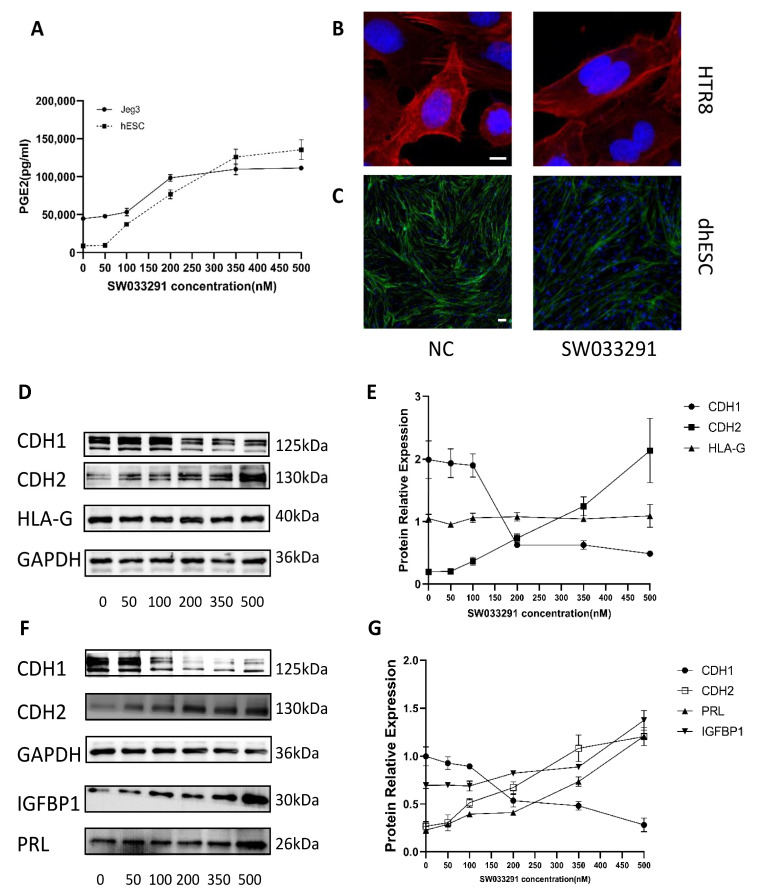
Function of 15-PGDH. (**A**) PGE2 levels in cell lysates were quantified by ELISA. The results are representative of at least three independent experiments. (**B**,**C**) F-actin staining revealed cytoskeletal changes in dhESCs and HTR8 cells in the NC (normal control) group and SW033291 group. (**D**,**E**) Western blot assays show the relative levels of EMT markers and HLA-G in Jeg3. (**F**,**G**) Western blot assays show the relative levels of EMT markers and decidualization markers in hESC. Each experiment was independently performed three times. CTB: cytotrophoblast; EVT: extravillous trophoblasts; EMT: epithelial–mesenchymal transition; DSC: decidual stromal cell. The experiment was repeated 3 times independently. Band intensities were quantified and normalized to the GAPDH values. Values are the mean ± SD.

**Figure 6 ijms-24-05111-f006:**
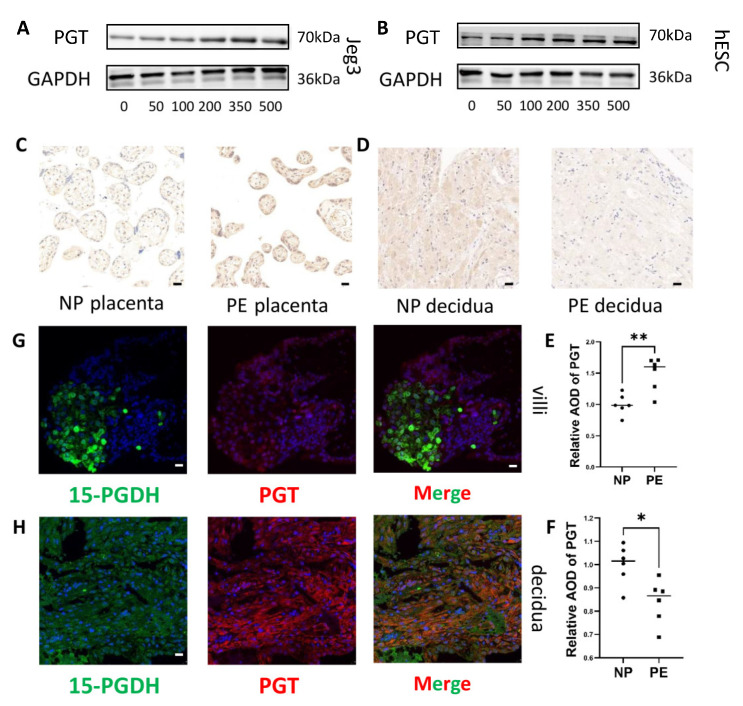
15-PGDH inhibitor increased PGT expression and PGT differential expression in PE patients. (**A**) Western blot assays showed that PGT expression increased in Jeg3 cells when treated with concentration of SW033291. (**B**) Western blot assays show that PGT expression increased in hESCs treated with concentration of SW033291. (**C**,**E**) Immunohistochemical staining showed the relative quantification of PGT in the placenta of the PE and NP groups (*n* = 6). Scale bar = 20 nm. (**D**,**F**) Immunohistochemical staining showed the relative quantification of PGT in the decidua of the PE and NP groups (*n* = 6). Scale bar = 20 nm. Values are the mean ± SD. * *p* < 0.05; ** *p* < 0.01. Each experiment was independently performed three times. (**G**,**H**) 15-PGDH and PGT, both located in trophoblasts and DSCs, were identified by immunofluorescence; red: PGT; green: 15-PGDH. G:Scale bar = 20 nm; H: Scale bar = 50 nm. The experiment was repeated 3 times independently.

**Figure 7 ijms-24-05111-f007:**
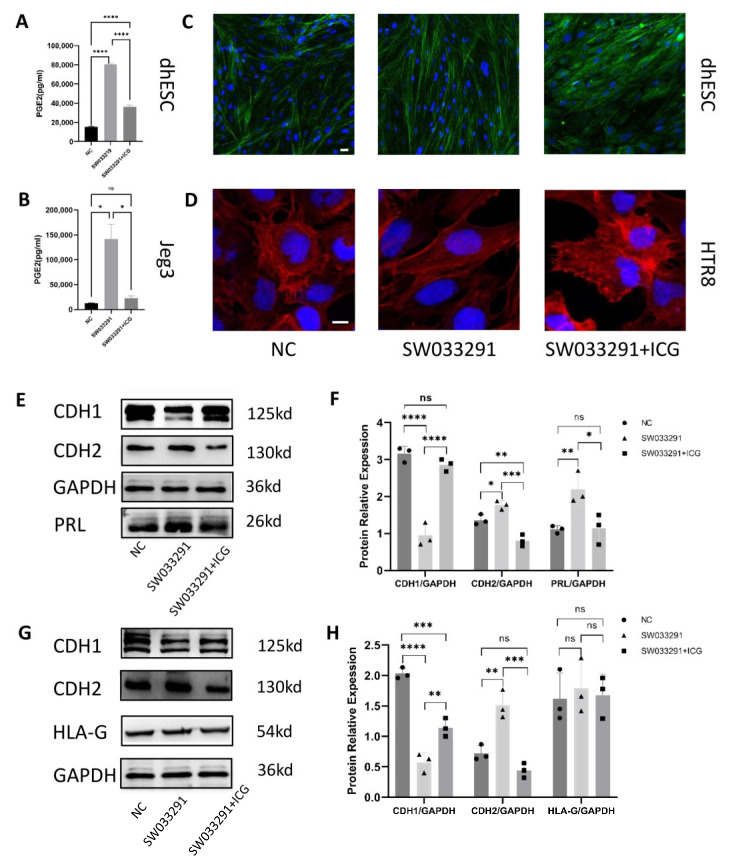
Function of the 15-PGDH inhibitor dependent on PGT. (**A**,**B**) PGE2 levels in cell lysates were quantified by ELISA. (**A**) represents PGE2 levels in dhESCs; (**B**) represents PGE2 levels in Jeg3 cells. (**C**,**D**) F-actin staining revealed cytoskeletal changes in dhESCs and HTR8 cells in the NC (normal control) group, SW033291 group, and SW033291+ICG group. Bar= 20 nm. (**E**,**F**) Western blot assays showed the relative levels of EMT markers and decidualization markers. (**G**,**H**) Western blot assays showed the relative levels of EMT markers and HLA-G. Each experiment was independently performed three times. GAPDH served as a loading control. Band intensities were quantified and normalized to the GAPDH values. The results are representative of at least three independent experiments. * *p* < 0.05; ** *p* < 0.01; *** *p* < 0.001; **** *p* < 0.0001; ns, not significant (one-way ANOVA). NC: normal control; ICG: indocyanine green; NP: normal pregnancy; PE: preeclampsia.

**Figure 8 ijms-24-05111-f008:**
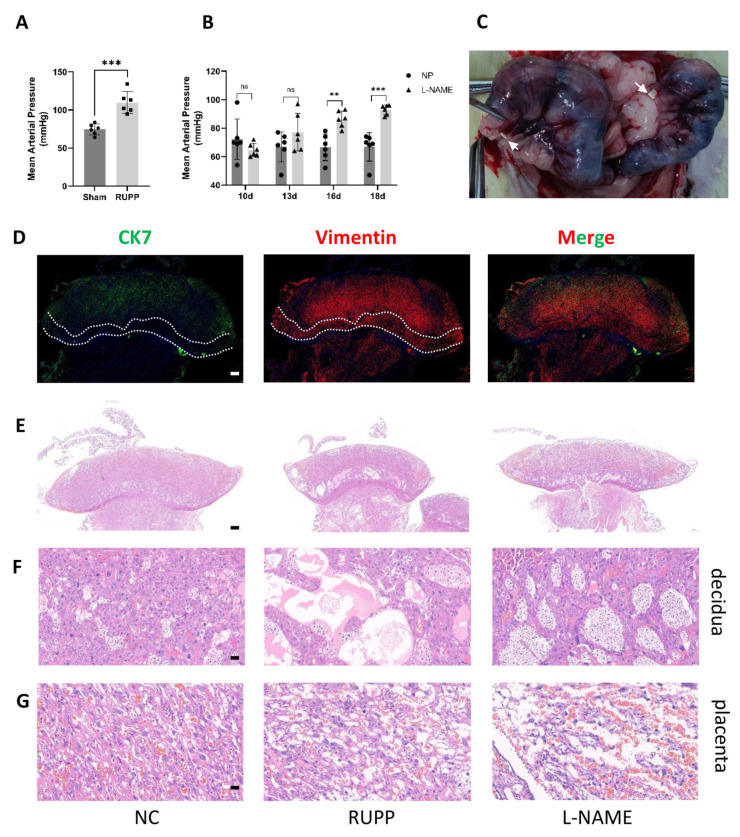
Preeclampsia animal model construction (**A**,**B**) The success of the animal model was confirmed by statistically analysing blood pressure, *n* = 6 for each group. ** *p* < 0.01; *** *p* < 0.001; ns, not significant. (**C**) Location of two silver clips on vessels between the ovary and uterus. (**D**) Determination of cell components of the maternal–feotal interface of SD rats. Green: CK7; red: vimentin, Bar = 500 nm. (**E**−**G**) HE staining showed the maternal–foetal interface structure change of RUPP and L-NAME rats vs. normal SD rats. *n* = 3 for each group, (**E**) shows the whole picture, scale bar = 500 nm, (**F**) shows the decidua, scale bar = 50 nm, and (**G**) shows the placenta, scale bar = 20 nm.

**Figure 9 ijms-24-05111-f009:**
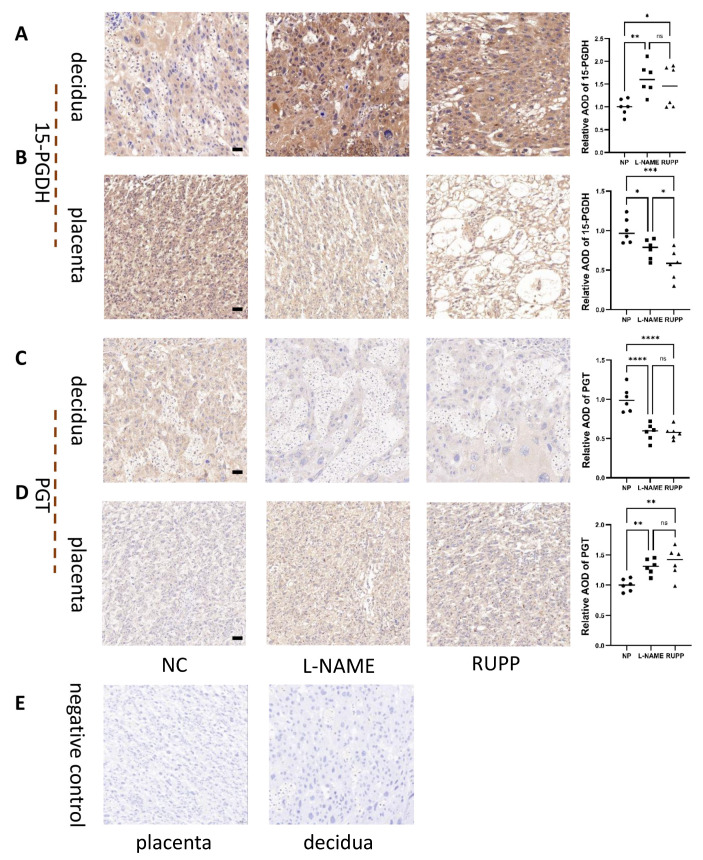
Enzyme 15-PGDH and PGT enzymes in the RUPP and L-NAME models. (**A**) Immunohistochemical staining showed the localization and relative quantification of 15-PGDH in the decidua of the RUPP and L-NAME rat groups. Brown staining represents the target protein, and (**B**) immunohistochemical staining showed the localization and relative quantification of 15-PGDH in the labyrinth zone of the RUPP and L-NAME vs. NP rat groups. *n* = 6 for each group. (**C**) Immunohistochemical staining showed the localization and relative quantification of PGT in the decidua of the RUPP and L-NAME rat groups. Brown staining represents the target protein, and (**D**) immunohistochemical staining showed the localization and relative quantification of PGT in the labyrinth zone of the RUPP and L-NAME vs. NP rat groups. The results are representative of at least three independent experiments. (**E**) represents negative control of placenta and decidua. Scale bar = 50 nm. * *p* < 0.05; ** *p* < 0.01; *** *p* < 0.001; **** *p* < 0.0001; ns, not significant (one-way ANOVA). *n* = 6 for each group. NP: normal pregnancy.

## Data Availability

All data associated with this study are available in the main text or the Appendix A.

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
