# Peer review of "The Enzyme 15-Hydroxyprostaglandin Dehydrogenase Inhibits a Shift to the Mesenchymal Pattern of Trophoblasts and Decidual Stromal Cells Accompanied by Prostaglandin Transporter in Preeclampsia"

_ijms, 2023, doi:10.3390/ijms24065111_

Round 1

Reviewer 1 Report

The study by Pang and colleagues determined whether  the enzyme 15-PGDH and prostaglandin trasporter, PGT are involved in preeclampsia focusing on the shared pathogenesis of fetal placenta and maternal decidua from the perspective of epithelial-mesenchymal transition (EMT)/mesenchymal-epithelial transition (MET) and explored the combined effects of 15-PGDH and PGT on the EMT/MET of trophoblasts and decidual stromal cells (DSCs).  Focus on link between decidualization and placental development is important but underexplored making the findings of the study warranted.  However, there are several points which should be addressed for publication.

1.      English language should be checked.

2.      The quality of Western blots should be improved.  There are numerous panels where multiple bands are observed but there is no mention why this is.  If there are variants of the proteins, this should be stated as well as state how these bands (single band, multiple bands) were quantitated for bar graphs which depict “protein relative expression.”  Please also indicate the molecular weights of the protein bands/include MW ladder or indication of protein size which corresponds to the bands being quantitated.

3.      Western blots in Supplemental data should show the molecular weight ladder or marker of molecular weights, blots should show entire blot or at least more than what is displayed, these figures do not show much more than what is in the manuscript text.

4.      For immunohistochemistry, negative controls should be included.

5.      For immunohistochemistry, there is no mention how data were quantitated (Relative IOD = ?).  Please provide additional detail beyond the use of ImageJ explaining data obtained for quantitation.

6.      In Figure 4, C (15-PGDH, CK7) please also include white triangle and  + sign signify in the figure legend.

7.      Please make the symbols for CDH1, CDH2, HLA-G, PRL and IGFBP1 larger in Figure 5 E and G.

8.      What was the sensitivity, intra- and inter-coefficients of variation for the PGE2 ELISA?

Reviewer 2 Report

This research is devoted to the study of the preeclampsia pathogenesis. Preeclampsia is the serious pathology of pregnancy. It is one of the most important causes of maternal and perinatal morbidity and mortality and has no tendency to decrease. This study is original and has undoubted scientific novelty. The introduction provides sufficient background and includes relevant references, but needs some additions. The design of this study is well considered. The research methods are well chosen and adequately described. The results are quite clearly presented, but the Figures need some corrections. All conclusions are supported by the results of the present study.

I have several comments about this manuscript:

1.      There are no references at the beginning of the introduction, although the authors write about "Previous studies".

2.      It should be added the relevance of preeclampsia study with references in the introduction.

3.      The authors should do serious work with the Figures. Firstly, some Figures are blurred, the inscriptions on them are poorly readable (for example, Fig.1G, 2B, 2D, 5E, 5G and others). Secondly, the photos have poor quality in the Fig.3A and 8D. The authors can change, for example, the photos’ resolution or zoom. Thirdly, some P-value are indicated on the graphs and mismatch with the description under the figure (for example, Fig. 4E, 6F: on the graphs P-value is ****, and there is no such sign in description under the figures). The arrangement of letters from C to G doesn’t correspond to the description under the Figure 1. In Fig. 6E and 6F the changes in the amount of PGT is the opposite and doesn’t correspond to the description in the text (line 222). What does "Sham" mean in Figure 8A?

The authors should check and correct all the Figures.

4.      The authors should to add a transcript of abbreviations where they occur for the first time in the text (for example, CDH1, CDH2 in line76, dhESC, udhESC in line 87, PRL in line 95 etc.).

5.      It is necessary to add references for methods in line 94.

6.      In line 119 "other immune cells" should be changed to "immune cells" or "other cells including immune cells" because DSCs and trophoblasts aren’t immune cells.

7.      English language and style are minor spell check required.

Round 2

Reviewer 1 Report

The authors have done a thorough job of revising the manuscript.  The only additional suggestion would be to define what the negative controls are for immunohistochemistry (for example, placental tissue without primary antibody).

Author Response

Dear Anonymous Reviewer:

We sincerely thank you for your kind work to revise our manuscript. And thank you very much for your valuable suggestion.

We have added what the negative controls are for immunohistochemistry in Materials and Methods, which is located in line 470. The revised portion is marked in red.

……Sections for negative control were incubated without primary antibody. ……

Reviewer 2 Report

The authors have made the necessary corrections to the manuscript. The article can be accepted without any changes.

Author Response

Dear Anonymous Reviewer:

We sincerely thank you for your kind work about our article.